# DTRNet: Precisely Correcting Selection Bias in Individual-Level Continuous Treatment Effect Estimation by Reweighted Disentangled Representation

**Mengxuan Hu**                                          *qtq7su@virginia.edu*
*School of Data Science*
*University of Virginia*

**Zhixuan Chu**                                          *zhixuanchu@zju.edu.cn*
*Zhejiang University*

**Sheng Li**                                             *shengli@virginia.edu*
*School of Data Science*
*University of Virginia*

**Reviewed on OpenReview:** *https://openreview.net/forum?id=1ZTfzA9bXw*

## Abstract

Estimating the individual-level continuous treatment effect holds significant practical importance in various decision-making domains, such as personalized healthcare and customized marketing. However, most current methods for individual treatment effect estimation are limited to discrete treatments and struggle to precisely adjust for selection bias under continuous settings, leading to inaccurate estimation. To address these challenges, we propose a novel Disentangled Representation Network (DTRNet) to estimate the individualized dose-response function (IDRF), which learns disentangled representations and precisely adjusts for selection bias. To the best of our knowledge, our work is the first attempt to precisely adjust for selection bias in continuous settings. Extensive results on synthetic and semi-synthetic datasets demonstrate that our DTRNet outperforms most state-of-the-art methods. Our code is available at DTRNet.

## 1 Introduction

In various fields, from medicine to marketing, estimating the causal effects of continuous treatments at individual level is not just an academic exercise; it is crucial for decision making. Take precision medicine as an example: The central question often focuses on determining the *"optimal dosage of medicine to achieve the optimal outcome for a given patient"*. Therefore, understanding the causal relationship between continuous treatments and outcomes can help develop customized medication regimens tailored to individual patients.

When estimating the effect of individual treatment (ITE), two predominant challenges arise: the inability to observe **counterfactual outcomes** and the presence of **selection bias**. For example, when a specific dose of treatment is assigned to a patient, only the factual outcome corresponding to that dose is observed, leaving counterfactual outcomes unobserved for other doses. Furthermore, unlike randomized controlled trials where treatments are assigned at random, the dosage a patient receives in practice may depend on certain patient-specific features (e.g., older individuals may receive higher dosages more frequently). This dependency can introduce selection bias, thereby compromising the accuracy of counterfactual outcome estimation. For example, it becomes challenging to accurately estimate the treatment effect of higher dosages in younger populations. Beyond conventional causal inference techniques such as stratification methods (Pearl, 2009) and matching methods (Abadie et al., 2004), recent research has harnessed representation learning

for counterfactual prediction and selection bias mitigation (Johansson et al., 2016; Shalit et al., 2017; Chu et al., 2021; 2022a;b; Schwab et al., 2020; Curth & van der Schaar, 2021; Bellot et al., 2022; Wang et al., 2022; Acharki et al., 2023; Huang et al., 2024). These approaches involve learning latent representations from covariates, balancing these representations across treatment groups to eliminate selection bias, and subsequently estimating counterfactual outcome on these balanced representations (Shalit et al., 2017).

Despite prior endeavors, several crucial challenges still remain unresolved. ❶ Most existing studies (Yao et al., 2021; Acharki et al., 2023; Huang et al., 2024) are limited to discrete treatment settings. These methods cannot be easily extended to continuous treatment settings due to the infinite number of unobserved counterfactual outcomes and the difficulty of adjusting for selection bias in an infinite set of treatments. ❷ Existing methods struggle to **precisely** adjust for selection bias. Many previous approaches resort to a simplistic and brute-force method, which balances the entire representations, leading to inaccurate estimation. However, we argue that not all the information in the latent representations should be balanced to adjust for selection bias (Greenland, 2008; Chu et al., 2020; 2023a). For example, although confounder factors in representations bring selection bias, they also contribute to outcome predictions (Hassanpour & Greiner, 2019a). Balancing instrumental factors in the representations is also theoretically implausible, as they are related to the treatment assignment and independent of all confounders (Wu et al., 2022); hence, they should not be identical across treatments. Attempting to balance them can lead to increased bias and variance in causal effect estimation (Myers et al., 2011). Therefore, the entire representations of input covariates should not be indiscriminately balanced.

Although some follow-up methods have attempted to address the first challenge by dividing continuous treatments into bins and assigning one network head to each bin (Schwab et al., 2020; Acharki et al., 2023), using a varying coefficient structure (Nie et al., 2021), or designing a specific loss for continuous treatments (Bellot et al., 2022; Zhu et al., 2024a), these methods do not guarantee any precise adjustment for selection bias (i.e., the second challenge). On the other hand, although Hassanpour & Greiner (2019b) demonstrated that disentangled representations can be leveraged to precisely correct for selection bias, it is based on the assumption of binary treatment and cannot be readily extended to continuous settings (i.e., the first challenge). To the best of our knowledge, there is no research that simultaneously solves these two problems, namely *generating appropriate disentangled representations that precisely adjust for selection bias to estimate the continuous treatment effect at the individual level, which is defined as the Individualized Dose-Response Function (IDRF).*

To address these challenges, we propose a novel method named Disentangled Representation Network (DTRNet). Specifically, we follow the convention assumption in causal inference (Wu et al., 2020; Kuang et al., 2017; Hassanpour & Greiner, 2019a) that covariates are determined by three latent factors: (1) Instrumental factors. (2) Confounder factors. (3) Adjustment factors. (as shown in Fig. 1 a). DTRNet first learns disentangled representations for each factor, providing the opportunity to precisely adjust for bias by using only the relevant representations instead of the entire representations. Then, we precisely adjust for selection bias by adopting a re-weighting function and predict outcomes based on the representations of confounder and adjustment factors through a varying coefficient network, which enables continuous treatment effect estimation. A rigorous theoretical proof supporting the debiasing ability of our re-weighting function is also provided. Our contributions are summarized as follows.

- We propose a novel method, DTRNet, which learns disentangled representations for unbiased continuous treatment effect estimation at the individual level. These disentangled representations provide a robust foundation for precisely adjusting for selection bias by using only the relevant representations rather than the entire set.

- The effectiveness of DTRNet's precise bias adjustment is supported by theoretical proofs and does not require prior knowledge of the treatment distribution, enhancing its practical applicability.

- We have conducted extensive experiments to validate the effectiveness and disentangling capability of our model. The results demonstrate that our method performs well on both synthetic and semi-synthetic datasets, with each component contributing to its advanced performance. Additionally, the results indicate that the learned disentangled components accurately capture the corresponding factors.

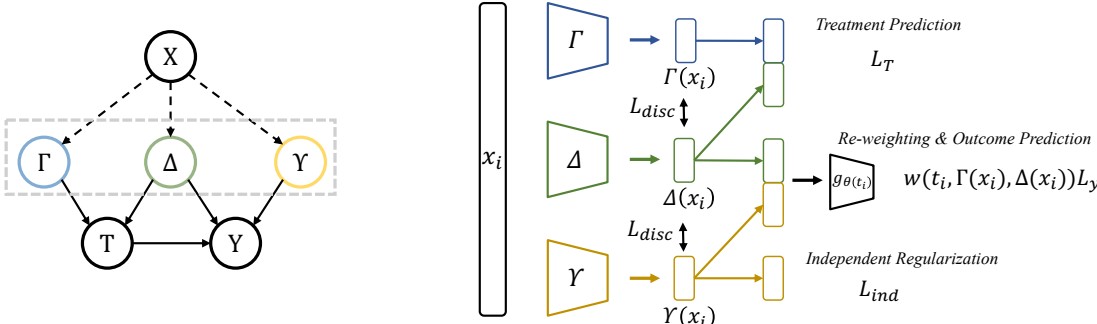

Figure 1: Causal graph and framework of our DTRNet. Figure (a) shows the causal graph involving covariates ($X$), treatment ($T$), outcome ($Y$), instrumental factors ($\Gamma$), confounder factors ($\Delta$), adjustment factors ($\Upsilon$). The solid line represents causal relations, and dot lines denote affiliations. Figure (b) shows the framework of DTRNet. Three contracted neural networks are utilized to obtain the deep representations of the three factors. Then $\Gamma(x_i)$ and $\Delta(x_i)$ are concatenated to predict the distribution of $t_i$. $\Delta(x_i)$, and $\Upsilon(x_i)$ are used to predict outcomes through a varying coefficient network $g_{\theta(t_i)}$, while $\Upsilon(x_i)$ attempts to encode little information about treatment.

## 2 Related Work

### 2.1 Potential Outcomes Framework

The potential outcomes framework, as proposed by (Rubin, 1974), is widely used to define the individual treatment effect (ITE). Specifically, to make the definition clear, we use binary treatments ($T = 1$ and $T = 0$) to illustrate, which can be further extended to multiple treatments by comparing their potential outcomes. For each individual $x_i$, there are two potential outcomes: $Y_i(T = 1)$ and $Y_i(T = 0)$ corresponding to two possible treatments ($T = 1$ and $T = 0$). As only one treatment can be taken for each individual, only one of the potential outcomes can be observed (observed outcome), and the remaining unobserved outcome is the counterfactual outcome. Therefore, one major challenge in estimating ITE is to infer the counterfactual outcome. After obtaining the counterfactual outcome, the ITE is calculated as follows.

$$ITE_i = Y_i(T = 1) - Y_i(T = 0). \tag{1}$$

Moreover, estimating the ITE is also challenged by the presence of selection bias. Some covariates may act as common causes for both the treatment and the outcome, leading to spurious correlations between them, which affect the accuracy of the treatment estimation.

### 2.2 Machine Learning for ITE Estimation

As mentioned above, selection bias and counterfactual prediction are two major challenges for ITE. To address these issues, various methods have been proposed (Alaa & van der Schaar, 2017; Hansen, 2008; Chipman et al., 2010; Hansen, 2008; Wager & Athey, 2018; Chu & Li, 2023; Yao et al., 2021; Acharki et al., 2023). Moreover, several state-of-the-art methods use deep representation learning models to estimate ITE based on treatment-invariant representations (Johansson et al., 2016; Shalit et al., 2017; Louizos et al., 2017; Yoon et al., 2018; Yao et al., 2018; Bellot et al., 2022; Wang et al., 2022; Acharki et al., 2023; Guo et al., 2023; Nagalapatti et al., 2024; Sui et al.; Huang et al., 2024; Zhu et al., 2024b; Chu et al., 2023b; 2024; Huang & Zhang, 2023; Zhu et al., 2022; Jiang et al., 2023; Gao et al., 2024; Pogodin et al., 2022). Specifically, the discrepancy loss between the deep representations of the treatment group and control group is used to balance the distribution of the two groups to adjust for selection bias. Subsequently, one network head for each treatment is built on the deep-balanced representations to estimate ITE. However, due to the commonly used two-head design in existing work, these models cannot be easily generalized to continuous treatment settings.

To address this limitation and estimate the average dose-response function (ADRF) under a continuous setting, several methods have been proposed (Schwab et al., 2020; Nie et al., 2021; Shi et al., 2019; Bellot et al., 2022; Jesson et al., 2022; Nagalapatti et al., 2024). Schwab et al. (Schwab et al., 2020) introduced a modification of TARNET (Shalit et al., 2017), called Dose Response Networks (DRNet), which divided continuous dosage into equally sized strata and assigned a head to each of them. To further achieve continuity of the ADRF, Nie et al. (Nie et al., 2021) proposed a varying coefficient neural network (VCNet). Instead of using a multi-head model, they used a varying coefficient prediction head whose weight depends on the treatment $t$, which improves the expressiveness of the treatment effect. In addition, generative adversarial net (GAN)-based models (Bica et al., 2020), transformer-based models (Zhang et al.), contrastive learning based models (Zhu et al., 2024a) have also been proposed to estimate the ADRF. However, the methods fail to accurately adjust for selection bias, as they do not make any adjustments to eliminate the bias, or resort to a simple and unsophisticated approach to balance the entire representations.

Hassanpour et al. Hassanpour & Greiner (2019a) proposed that disentangled representations of covariates enhance the capture of their underlying factors, thereby improving ITE estimation performance. Instead of balancing the entire deep representations, they only applied discrepancy loss on the representations of adjustment factors and used re-weighting methods to adjust for selection bias brought by confounders. As a result, no confounding variables are discarded. However, the method can only work with binary treatments. While subsequent research efforts (Curth & van der Schaar, 2021; Chauhan et al., 2023) have introduced more refined disentanglement representations, it is worth noting that these advancements are predominantly tailored to binary settings.

Unlike existing work, our DTRNet attempts to address the two limitations mentioned above by generating appropriate disentangled representations for three underlying factors and precisely adjusting for selection bias to estimate IDRF.

## 3 Methodology

In this section, we begin by defining the problem and providing an overview of DTRNet. Next, we delve into the functions and explanations of each component of our DTRNet. Following this, we provide theoretical substantiation that our devised re-weighting function is able to precisely adjust for selection bias. Lastly, we discuss the advantages of our approach in comparison to other state-of-the-art models.

### 3.1 Problem Setting

Let $\mathcal{D} = \{x_i, y_i, t_i\}_{i=1}^N$ denote a dataset of size $N$, where $x_i, y_i, t_i$ are independent realisations of random variables $X, Y, T$ with support $(\mathcal{X}, \mathcal{Y}, \mathcal{T} = [0,1])$, respectively. We refer to $X \in \mathbb{R}^m$ as covariates, which contain information about features of an unit. $Y$ represents the outcome, and $T$ represents the continuous treatment in the range from 0 to 1 (Nie et al., 2021). Our goal is to estimate the Individualized Dose-Response Function (IDRF) with continuous treatments by eliminating selection bias.

**Definition 3.1.** (Individualized Dose-Response Function (IDRF)). IDRF measures the treatment effect for an individual under continuous treatments, which can be defined as:

$$\mu(t, x) = \mathbb{E}[Y(T = t)|X = x, T = t] \tag{2}$$

We assume that covariates are generated from three underlying factors: (1) *instrumental factors* ($\Gamma(x)$) that are associated with the treatment but not with the outcome except through the treatment, (2) *confounder factors* ($\Delta(x)$) that are associated with both the treatment and outcome, and (3) *adjustment factors* ($\Upsilon(x)$) that are predictive of the outcome but not associated with the treatment. Therefore, the treatment assignment is affected by instrumental factors and confounder factors, while the outcome is affected by confounder factors and adjustment factors. This assumption is commonly used in previous research to disentangle the covariates for precise information extraction (Yao et al., 2021; Wu et al., 2020), an illustrative example can be found in Appendix I. The underlying relationship can be illustrated using a causal graph shown in Fig. 1(a). It is important to note that IDRF identification is made under the following convention assumptions (Wu et al., 2020; Kuang et al., 2017; Hassanpour & Greiner, 2019a).

**Assumption 3.2** (Stable Unit Treatment Value Assumption (SUTVA))**.** There are no interactions between units, and there is only one version of each treatment, that is, if two different units $i$ and $j$ have the same value for their treatment variable, then they receive the same treatment.

**Assumption 3.3** (Ignorability)**.** The potential outcome $Y(T = t)$ is independent of the treatment assignment given all covariates. Formally, $Y(T = t) \perp\!\!\!\perp T|X$.

**Assumption 3.4** (Positivity)**.** Every unit should have non-zero probabilities to be assigned in each treatment group. Formally, $\mathbb{P}(T = t|X = x) \neq 0, \forall t \in \mathcal{T}, \forall x \in X$.

**Assumption 3.5** (Generation of Covariates)**.** Given a set of covariates, denoted as X, we assume X follows the joint distribution of instrumental factors $\Gamma$, confounder factors $\Delta$, and adjustment factors $\Upsilon$. Formally, $\mathbb{P}(X) = \mathbb{P}(\Gamma, \Delta, \Upsilon)$.

### 3.2 DTRNet Model

DTRNet is designed according to the causal graph in Fig. 1. It first learns disentangled representations of covariates and subsequently estimates $T$ and $Y$ based on the corresponding representations. Finally, it corrects for selection bias by utilizing relevant representations instead of the entire set. In particular, three contracted feedforward neural networks are utilized to obtain disentangled representations of three factors $\{\Gamma(x_i), \Delta(x_i), \Upsilon(x_i)\}$ defined in Section 3.1. Then $\Gamma(x_i)$ and $\Delta(x_i)$ are concatenated to predict the distribution of $t_i$ using a conditional density estimator $\mathbb{P}(t_i|\Gamma(x_i), \Delta(x_i))$. $\Delta(x_i)$ and $\Upsilon(x_i)$ are used to predict the final outcome through a varying coefficient network $g_{\theta(t_i)}(\Delta(x_i), \Upsilon(x_i))$, while $\Upsilon(x_i)$ attempts to encode little information about treatment. Typically, a re-weighting function is responsible for precisely adjusting for selection bias. The DTRNet framework is shown in Fig. 1(b), with the objective function defined as follows:

$$J(X, T, Y) = \frac{1}{N} \sum_{i=1}^{N} (w(t_i, \Gamma(x_i), \Delta(x_i))L_y + \alpha L_T + \gamma L_{ind} + \lambda L_{reg}) + \beta L_{disc}, \tag{3}$$

where $w(t_i, \Gamma(x_i), \Delta(x_i))$ denotes the re-weighting function to mitigate selection bias; $L_y$ and $L_T$ are the prediction losses for outcome $Y$ and treatment $T$, respectively. $L_{disc}$ quantifies discrepancies between latent representations $(\Gamma(x_i), \Delta(x_i), \Upsilon(x_i))$; $L_{ind}$ promotes the independence between $\Upsilon(x_i)$ and treatment $t_i$; while $L_{reg}$ serves as a regularization term against overfitting. $\alpha$, $\beta$, $\gamma$, and $\lambda$ are hyperparameters balancing the different terms in the objective function. In the following, we present the details of each term.

**Factual Loss.** Factual loss is used to force $\Delta(x_i)$ and $\Upsilon(x_i)$ to extract more predictive information from the covariates. As three arrows pointing to $Y$ in Fig. 1(a), we aim to estimate the factual outcome $y_i$ from three input, $\Delta(x_i)$, $\Upsilon(x_i)$, and assigned treatment $t_i$ for unit $i$. To simultaneously preserve the influence of treatment and maintain the continuity of the dose-response curve, we adopt a varying coefficient neural network $g_{\theta(t_i)}$ to estimate outcomes (Nie et al., 2021). Factual loss is computed by comparing the ground truth $y_i$ with our estimated value:

$$L_y = L\left(y_i, g_{\theta(t_i)}(\Delta(x_i), \Upsilon(x_i))\right), \tag{4}$$

where $L$ represents the loss function and we use the mean square error in our paper. The varying coefficient structure utilizes a function $g_{\theta(t)}$ with varying parameters $\theta(t)$ instead of fixed parameters to predict outcomes. By leveraging this structure, continuous treatment effect can be estimated by incorporating continuous $t$ in outcome estimation. Especially, a B-spline of degree $p$ with $q$ knots, resulting in $k = p + q + 1$ basis, is used to model $\theta(t)$. Let $\mathbf{B} = [b_1, b_2, ..., b_k] \in \mathbb{R}^{n \times k}$ denote the spline basis for the treatment $T \in \mathbb{R}^{n \times 1}$. For a single-layer feedforward network with $m$ inputs and $n$ outputs, the function is given by $f_{\theta(t)} = \sum_{i=1}^{k}(b_i \cdot (\mathbf{XW}))$ , where $\mathbf{W} \in \mathbb{R}^{m \times n \times k}$ is the optimizable weight.

**Treatment Loss.** We incorporate a treatment loss in our model to enhance the encoding of $\Gamma(x_i)$ and $\Delta(x_i)$ with respect to $t_i$. Prior studies have usually predicted the probability of treatment using the full covariate representations (Shi et al., 2019; Nie et al., 2021). This approach can inadvertently leverage irrelevant information, such as adjustment factors, from the representations. To mitigate this, we estimate

the probability of $t_i$ from the concatenation of $\Gamma(x_i)$ and $\Delta(x_i)$ using a conditional density estimator $\pi(t|\Gamma(x_i), \Delta(x_i))$, yielding more accurate treatment predictions. In this paper we adopt a naive density estimator (Nie et al., 2021), which approximates the conditional density by dividing $t \in [0, 1]$ equally into $B$ intervals, and estimating the conditional density $\pi(t|\Gamma(x_i), \Delta(x_i))$ on the $B + 1$ grid points using a simple neural network $\pi^{NN}(\Gamma(x_i), \Delta(x_i)) = softmax(\mathbf{W}_t \cdot concat(\Gamma(x_i), \Delta(x_i))) \in \mathbb{R}^{B+1}$, where $\mathbf{W}_t$ is the parameter for the network, and the densities for other values of $t$ are derived via linear interpolation. Performance is measured using a negative logarithmic likelihood loss.

$$L_T = -\log[\mathbb{P}(t_i|\Gamma(x_i), \Delta(x_i))]. \tag{5}$$

**Discrepancy Loss.** The discrepancy loss is employed to enhance the separation of three distinct representations, namely $\Gamma(x)$, $\Delta(x)$, and $\Upsilon(x)$. This ensures that each representation encodes only its specific information, thereby strengthening disentanglement in the latent space. The formula for this loss is presented as follows:

$$L_{disc} = \frac{1}{L_D(\Gamma(x), \Delta(x)) + L_D(\Delta(x), \Upsilon(x))}, \tag{6}$$

where

$$L_D(\Gamma(x), \Delta(x)) = \frac{1}{N} \sum_i^N \sum^D \Delta(x_i) \log\left(\frac{\Delta(x_i)}{\Gamma(x_i)}\right) = \frac{1}{N} \sum_i^N \sum^D \Delta(x_i)(\log(\Delta(x_i)) - \log(\Gamma(x_i))), \tag{7}$$

where $D$ is the dimension of the latent embedding, $L_D(\Gamma(x), \Delta(x))$ denotes the average divergence loss between $\Gamma(x)$ and $\Delta(x)$ across the units inspired by the KL divergence. If $\Gamma(x_i)$ and $\Delta(x_i)$ are identical for all unites, the result will be 0. On the contrary, if $\Gamma(x_i)$ and $\Delta(x_i)$ are distinctly different for all units, the value will be higher. Therefore, through the collaboration of other modules, $L_{disc}$ encourages all disentangled representations to encode only the relevant information. The definition of $L_D$ and additional details can be found in Appendix E.

**Independent Loss.** Independent loss is specifically applied to balance the $\Upsilon(x_i)$ instead of the entire representations, ensuring that the learned factors $\Upsilon(x_i)$ do not contain any information about $t_i$ and that all information related to $t_i$ is encoded in $\Gamma(x_i)$ and $\Delta(x_i)$. Previous studies (Shalit et al., 2017; Johansson et al., 2016; Hassanpour & Greiner, 2019a;b; Acharki et al., 2023; Guo et al., 2023) have emphasized the need to balance adjustment representations for the treatment group and control group (binary treatment setting), most of them done by inducing the desired independence using a discrepancy loss. However, within the framework of continuous treatments, the endeavor to achieve balance adjustment representations for every value $t_i$ becomes infeasible. Hence, we intend to push $\Upsilon(x_i)$ to embed little information about the treatment by forcing the performance of the treatment probability estimation from adjustment representation to be poor, which motivates us to minimize the following "positive" log-likelihood loss:

$$L_{ind} = \log(\mathbb{P}(t_i|\Upsilon(x_i)). \tag{8}$$

In particular, this independent loss allows us to encode all information of $t_i$ in $\Gamma(x_i)$ and $\Delta(x_i)$ instead of $\Upsilon(x_i)$, which facilitates the precisely adjustment for selection bias through the re-weighting function (discussed in next paragraph). This is one of the key contributions of our paper. Furthermore, compared to previous studies that balance the entire representations of covariates, our approach does not balance confounder factors since they contain valuable information about outcome prediction (Hassanpour & Greiner, 2019b). Moreover, we also exclude instrumental factors since they are related to the treatment assignment and independent of all confounders (Wu et al., 2022). Hence, they should not be balanced.

**Re-weighting Function.** Recall one of our objectives is to precisely eliminate selection bias. Inspired by (Imbens, 2000; Kloek & Van Dijk, 1978), we derive "propensity score" $\mathbb{P}(t_i|\Gamma(x_i), \Delta(x_i))$ from $\Gamma(x_i)$ and $\Delta(x_i)$ and use the inverse of it to re-weight the prediction loss of outcomes as follows:

$$w(t_i, \Gamma(x_i), \Delta(x_i)) = \frac{1}{\mathbb{P}(t_i|\Gamma(x_i), \Delta(x_i))}, \tag{9}$$

Table 1: Performance comparison between DTRNet and baselines. Numbers reported are MISE/AMSE± standard deviation) on Synthetic, IHDP, and News with 50 runs.

| Method | Synthetic Data | | News | | IHDP | |
|---|---|---|---|---|---|---|
| | MISE | AMSE | MISE | AMSE | MISE | AMSE |
| Dragonet | $0.1854 \pm 0.0232$ | $0.0415 \pm 0.0081$ | $1.3241 \pm 0.1617$ | $0.0535 \pm 0.0053$ | $4.7034 \pm 0.5860$ | $0.9549 \pm 0.3005$ |
| Dragonet_TR | $0.1720 \pm 0.0219$ | $0.0281 \pm 0.0095$ | $1.3147 \pm 0.1594$ | $0.0401 \pm 0.0062$ | $4.2877 \pm 0.4226$ | $0.6490 \pm 0.1660$ |
| DRNet | $0.1849 \pm 0.0232$ | $0.0409 \pm 0.0081$ | $1.3248 \pm 0.1616$ | $0.0542 \pm 0.0054$ | $4.7394 \pm 0.6036$ | $0.9581 \pm 0.3324$ |
| DRNet_TR | $0.1752 \pm 0.0334$ | $0.0315 \pm 0.0235$ | $1.3148 \pm 0.1601$ | $0.0403 \pm 0.0060$ | $4.1313 \pm 0.6320$ | $0.6140 \pm 0.1954$ |
| VCNet | $0.1545 \pm 0.0248$ | $0.0173 \pm 0.0093$ | $2.3372 \pm 0.1808$ | $0.0384 \pm 0.0367$ | $3.6651 \pm 0.6409$ | $0.6755 \pm 0.4875$ |
| VCNet_TR | $0.1418 \pm 0.0299$ | $0.0142 \pm 0.0072$ | $2.3289 \pm 0.2009$ | $0.0378 \pm 0.0401$ | $3.7935 \pm 1.3625$ | $1.2302 \pm 1.2198$ |
| TransTEE | $0.2033 \pm 0.0978$ | $0.0552 \pm 0.0884$ | $\mathbf{1.2849 \pm 0.1587}$ | $0.0153 \pm 0.0066$ | $4.1562 \pm 0.8053$ | $1.8529 \pm 1.1155$ |
| CIRCE | $0.6854 \pm 0.4050$ | $0.5406 \pm 0.4040$ | $1.7667 \pm 0.1995$ | $0.6030 \pm 0.0291$ | $10.4413 \pm 5.7308$ | $8.218 \pm 5.4327$ |
| DTRNet (Ours) | $\mathbf{0.1414 \pm 0.0256}$ | $\mathbf{0.0131 \pm 0.0072}$ | $1.7846 \pm 0.2202$ | $\mathbf{0.0104 \pm 0.0044}$ | $\mathbf{3.5376 \pm 0.5285}$ | $\mathbf{0.4254 \pm 0.3710}$ |

where $\mathbb{P}(t_i|\Gamma(x_i), \Delta(x_i))$ is the direct output of the conditional density estimator for treatment. Hence, it does not require additional computation or prior knowledge about the treatment distribution as in (Hassanpour & Greiner, 2019b). Furthermore, we can precisely remove bias attributable to the confounder and instrumental factors instead of the unrelated part ($\Upsilon(x_i)$). A detailed proof follows.

### 3.3 Theoretical Proof of Bias Elimination

In this section, we outline the theoretical derivation of our re-weighting function, inspired by the importance sampling theory (Hassanpour & Greiner, 2019a; Kloek & Van Dijk, 1978). We then provide proofs for its debiasing ability.

**Definition 3.6.** Let $\Delta, \Upsilon : \mathcal{X} \to \mathcal{R}$ be the representation functions for confounder factors and adjustment factors, respectively. Let $g_{\theta(t)} : \mathcal{R} \times \mathcal{R} \times \mathcal{T} \to \mathcal{Y}$ be the prediction function defined in the representation space $\mathcal{R} \times \mathcal{R}$. The expected loss for the unit and treatment pair $(x, t)$ is:

$$l_{\Delta, \Upsilon, g_{\theta(t)}}(x, t) = \int_{\mathcal{Y}} L(Y(t), g_{\theta(t)}(\Delta(x), \Upsilon(x)))\mathbb{P}(Y(t)|x)dY(t). \tag{10}$$

**Definition 3.7.** The expected unbiased IDRF loss across all treatment $t \in T$ is:

$$\epsilon = \mathbb{E}_x\left[\int_{\mathcal{T}} l_{\Delta, \Upsilon, g_{\theta(t)}}(x, t)dt\right] = \int_{\mathcal{X}} \int_{\mathcal{T}} l_{\Delta, \Upsilon, g_{\theta(t)}}(x, t)\mathbb{P}(x)dtdx. \tag{11}$$

**Lemma 3.8.** *(Importance Sampling (Kloek & Van Dijk, 1978)) Let $p(x)$ be a probability density for a random variable $X$ defined on $\mathbb{R}^d$, then for any probability density $q(x) \in \mathbb{R}^d$ that satisfies $q(x) > 0$ whenever $f(x)p(x) \neq 0$, we have:*

$$\mathbb{E}_{x \sim p(x)}[f(x)] = \mathbb{E}_{x \sim q(x)}\left[f(x)\frac{p(x)}{q(x)}\right].$$

The Lemma suggests that importance sampling facilitates the computation of the expectation of a target function $f(x)$ under an unknown distribution $p(x)$ by weighting the function with $\frac{p(x)}{q(x)}$ under a known distribution $q(x)$.

**Theorem 3.9.** *Let $p(x, t')$, $p(x, t)$ denote the counterfactual and factual probability density for unit $x$, respectively. Under the conditions of the lemma 3.8, the expected loss functions for factual and counterfactual outcomes are:*

$$\epsilon_F = \mathbb{E}_{x, t \sim p(x, t)}[l_{\Delta, \Upsilon, g_{\theta(t)}}(x, t)], \tag{12}$$

$$\epsilon_{CF} = \mathbb{E}_{x, t \sim p(x, t')}[l_{\Delta, \Upsilon, g_{\theta(t)}}(x, t)] = \mathbb{E}_{x, t \sim p(x, t)}\left[l_{\Delta, \Upsilon, g_{\theta(t)}}(x, t)\frac{p(x, t')}{p(x, t)}\right], \tag{13}$$

*where $t'$ represents all counterfactual treatments of $x$, especially $t' = \{t'|t' \in \mathcal{T}, t' \neq t\}$.*

Table 2: Results of ablation study. The top row shows the average MISE and AMSE of DTRNet across 50 runs for the three datasets. Subsequent rows present performance when the respective component is disabled. $\Delta$ values in parentheses represent the percentage change in AMSE and MISE relative to the top row results.

| Method | Synthetic Data | | News | | IHDP | |
|---|---|---|---|---|---|---|
| | MISE | AMSE | MISE | AMSE | MISE | AMSE |
| original | 0.1414 | 0.0131 | 1.7846 | 0.0104 | 3.5376 | 0.4254 |
| alpha | 0.1413($\Delta$ 0.0% ↑) | 0.0132($\Delta$ 0.8% ↑) | 2.6400($\Delta$ 47.9% ↑) | 0.0291($\Delta$ 179.8% ↑) | 3.606($\Delta$ 1.9% ↑) | 0.5115($\Delta$ 39.2% ↑) |
| beta | 0.1414($\Delta$ 0.0% ↑) | 0.0137($\Delta$ 4.6% ↑) | 2.6547($\Delta$ 48.8% ↑) | 0.0319($\Delta$ 206.7% ↑) | 3.7856($\Delta$ 7.0% ↑) | 0.6310($\Delta$ 48.3% ↑) |
| gamma | 0.1411 ($\Delta$ 0.2% ↓) | 0.0131($\Delta$ 0.0% ↑) | 2.6871($\Delta$ 50.5% ↑) | 0.0297($\Delta$ 185.5% ↑) | 3.5951($\Delta$ 2.0% ↑) | 0.4955($\Delta$ 16.5% ↑) |
| re-weighting | 0.1640($\Delta$ 16.0% ↑) | 0.0212($\Delta$ 61.8% ↑) | 2.2790($\Delta$ 27.7% ↑) | 0.0136($\Delta$ 30.7% ↑) | 3.6804($\Delta$ 4.0% ↑) | 0.9521($\Delta$ 123.8% ↑) |

The theorem implies that the expected loss function for counterfactual outcomes can be derived by re-weighting the factual loss. Hence, we can cooperate the counterfactual loss with the factual loss and optimise them together through a designed weight; $w = 1 + \frac{\mathbb{P}(x,t')}{\mathbb{P}(x,t)} = \frac{1}{\mathbb{P}(t|\Gamma(x),\Delta(x))}$. For further details, please refer to Appendix A.

**Theorem 3.10.** *(Bias Removal with Weighted Loss) Under the theorem 3.9 and all assumptions, we have:*

$$\mathbb{E}_{x,t \sim p(x,t)}[w \cdot l_{\Delta,\Upsilon,g_{\theta(t)}}(x,t)] = \epsilon.$$

This theorem states that the weighted loss is an unbiased estimation of IDRF loss, indicating our re-weighting function is advanced in eliminating selection bias. See Appendix A for a detailed proof.

### 3.4 Discussions

Our proposed DTRNet is built upon existing works such as (Hassanpour & Greiner, 2019b) and (Nie et al., 2021). However, our approach extends the existing works in several significant ways. (1) The disentangled method in (Hassanpour & Greiner, 2019b) can only provide a causal effect under binary treatments, while our method facilitates a mixed type of treatments, including continuous treatments. (2) We introduce a novel design for the independent loss. Instead of simply and brutally minimizing the discrepancy between the entire treatment representations and the control representations, we minimize the amount of treatment information that can be extracted from adjustment factors, which facilitates the precise adjustment for selection bias in re-weighting function. (3) We use the direct output of the conditional density estimator for re-weighting, which helps precisely eliminate bias and does not require additional computation or prior knowledge about the treatment distribution. Furthermore, in the ablation study, we demonstrate the importance of this component for model performance. (4) We provide rigorous theoretical proofs substantiating the debiasing ability of the re-weighting function. Finally, our DTRNet employs a discrepancy loss between different deep representations, which ensures that each representation only encodes the relevant information. Overall, DTRNet offers an extensive ability to generate disentangled representations on which selection bias can be precisely adjusted.

## 4 Experiments

In this section, we present extensive experimental results on three datasets to demonstrate the effectiveness of DTRNet and address the following three research questions: **Q1:** How effective is DTRNet in estimating IDRF and adjusting for selection bias compared to the state-of-the-art methods? **Q2:** What is the individual contribution of each component in our model, including the treatment loss, discrepancy loss, independent loss, and re-weighting function? **Q3:** Can deep disentangled representations be effective in identifying the three underlying factors?

### 4.1 Datasets and Baselines

As the ground truth of treatment effects is often unknown in practice, especially for continuous treatments, existing studies mainly rely on synthetic datasets and semi-synthetic datasets that manually construct

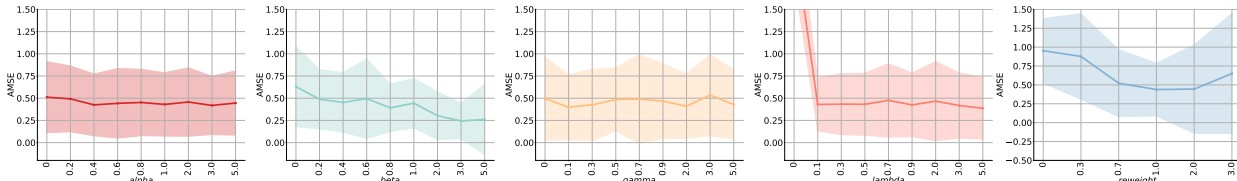

Figure 2: MISE sensitivity analysis with different values of $\alpha$, $\beta$, $\gamma$, $\lambda$ and different proportions of re-weighting value on 50 repeats of IHDP datasets. The standard deviation band is also plotted.

Figure 3: AMSE sensitivity analysis with different values of $\alpha$, $\beta$, $\gamma$, $\lambda$ and different proportions of re-weighting value on 50 repeats of IHDP datasets. The standard deviation band is also plotted.

treatments and outcomes given real-world features (Curth et al., 2021; Wang et al., 2022; Bellot et al., 2022; Nie et al., 2021). We follow this convention to build one synthetic dataset and two semi-synthetic datasets: News[1] and IHDP (Hill, 2011) for evaluation. To evaluate the effectiveness of selection bias adjustment, **we intentionally design all training sets to contain selection bias, while test sets are unbiased.** Hence, if a model is trained on a biased training set and performs well on the corresponding test set, it provides evidence of the model's ability to eliminate bias. We generate the three datasets following (Nie et al., 2021). Details can be found in Appendix G. We evaluate the performance of our proposed DTRNet against several state-of-the-art methods for IDRF estimation, including Dragonet, Dragonet_TR (Shi et al., 2019), DRNet, DRNet_TR (Schwab et al., 2020), VCNet, VCNet_TR (Nie et al., 2021), and TransEE (Zhang et al.), where TR refers to targeted regularization. The details of baselines are as follows.

- **Dragonet**: (Shi et al., 2019) used a three-headed architecture to predict the propensity score and conditional outcome from covariates and treatment information. The model was later improved by(Nie et al., 2021) by using separate heads for treatment in different intervals to adjust for continuous treatments.

- **DRNet**: (Schwab et al., 2020) proposed to divide continuous treatments into several intervals and assign one head to each interval to generate the dose-respond curve. Following (Nie et al., 2021), DRNet was improved by adding a conditional density estimation head for treatment estimation.

- **Vcnet**: Nie et al. (2021) introduced a varying coefficient structure to allow the prediction head parameters to be functions of continuous treatments.

- **TransTEE**: (Zhang et al.) adopted the transformer backbones to estimate treatment effect.

- **CIRCE**: (Pogodin et al., 2022) introduces CIRCE as a regularizer in settings where $X$ is used to predict $Y$. Specifically, the regularizer aims to learn neural features $\phi(X)$ of the data $X$ such that $\phi(X)$ is independent of some metadata $Z$ given $Y$.

We also include non-neural network based models: Causal Forest (Wager & Athey, 2018), Bart (Chipman et al., 2010) and GPS (Imbens, 2000), due to the space limitation, we show the performance in the Appendix D.

---

[1]https://archive.ics.uci.edu/ml/datasets/bag+of+words

## 4.2 Implementation Details

All the neural network-based methods are trained for 800 epochs with the SGD optimizer (momentum = 0.9). To mitigate the risk of overfitting or underfitting, we apply an early stop technique. For the three deep representation networks in our model, we implement them as fully connected networks with two hidden layers, and each layer has 50 hidden units using ReLU activation. We also use two-hidden-layer settings (each with 50 hidden units) for the $Y$ prediction network. The choice of the model structure aligns with previous approaches (Zhang et al.; Nie et al., 2021). We use grid search tuning to tune the following hyperparameters: $\alpha, \beta, \gamma \in \{0.1, 0.2, 0.4, 0.6\}$. The choice of this range is based on the common practice of weighting loss terms between 0 and 1. Specifically, in our paper, these hyperparameters refer to the weighting of each loss term, except for the factual loss (the loss between $y_i$ and $\hat{y}_i$). Since the factual loss represents the prediction $y$, which is a crucial component of the total loss, assigning a high weight to other terms could diminish the impact of the factual loss. Therefore, we chose the range $\{0.1, 0.2, 0.4, 0.6\}$, also considering computational and time costs. Typically, we choose the learning rate $lr \in \{0.0001, 0.00005, 0.00001\}$. For other hyperparameters, e.g., the number of knots and the degree of B-spline, we follow the setting of (Nie et al., 2021) which is also tuned on the same configurations of datasets. For each dataset, we generate 50 runs for training and 20 runs for tuning the aforementioned hyperparameters. The best hyperparameter settings for our method as well as the baseline can be found in Appendix F.

## 4.3 Results and Analysis

To answer **Q1**, we follow previous works (Zhang et al.; Nie et al., 2021; Bellot et al., 2022; Zhu et al., 2024a) to report the mean integrated squared error (MISE)

$$MISE = \int_{\mathcal{X}} [\int_{\mathcal{T}} (\hat{y}_i(t) - y_i(t))^2 dt] \mathbb{P}(x) dx, \tag{14}$$

and the mean squared error (AMSE). More details can be found in Appendix B.

$$AMSE = \int_{\mathcal{T}} [\frac{1}{n} \sum_{i=1}^{n} (\hat{y}_i(t) - y_i(t))]^2 \mathbb{P}(t) dt, \tag{15}$$

to evaluate the performance of the models in estimating the individual level and the population level dose response curve, respectively. Since the integral of $t$ is intractable, we apply all $t$ values existing in the current datasets on each unit to approximate the MISE and AMSE, that is,

$$M\hat{I}SE = \frac{1}{n} \frac{1}{|\mathcal{T}|} \sum_{i=1}^{n} \sum_{t \in \mathcal{T}} (\hat{y}_i(t) - y_i(t))^2, \tag{16}$$

$$A\hat{M}SE = \frac{1}{|\mathcal{T}|} \sum_{t \in \mathcal{T}} [\frac{1}{n} \sum_{i=1}^{n} (\hat{y}_i(t) - y_i(t))]^2, \tag{17}$$

where $|\mathcal{T}|$ is the number of different treatment values. To ensure fair and reliable comparisons, we evaluate the performance of all models on 50 repetitions of three different datasets and report the mean and standard deviation of the MISE and AMSE. As presented in Table 1, DTRNet consistently outperforms the majority of baselines across all datasets, achieving satisfactory MISE and the lowest AMSE while demonstrating commendable stable performance. Specifically, the less satisfactory MISE performance on the News dataset is attributable to the distinct data generation process, where all features act as confounders. This does not align with real-world scenarios or our study assumptions, which is why the performance is slightly lower compared to some of the baselines. These results demonstrate not only the effectiveness of DTRNet in IDRF estimation but also its ability to adjust for selection bias.

## 4.4 Ablation Study

To answer **Q2**, we conduct several ablation studies to evaluate the three major components of our model, including the treatment loss $L_T$, discrepancy loss $L_{disc}$, independent loss $L_{ind}$, and the re-weighting function

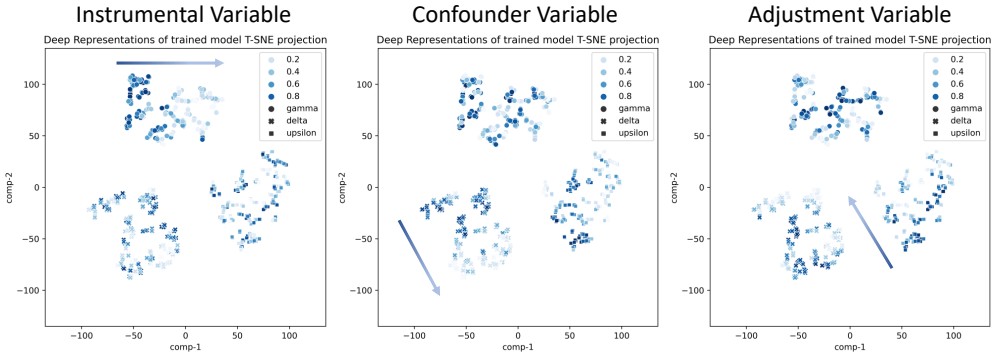

Figure 4: t-SNE plots of the three deep representations with respect to the instrumental variable, confounder variable, and adjustment variable. Point shapes represents the three types of deep representation. Point colors represents the value of the corresponding variable. Typically, arrows in the figures are manually added to provide a clear visual indication of the directions in which the color changes in the corresponding representations.

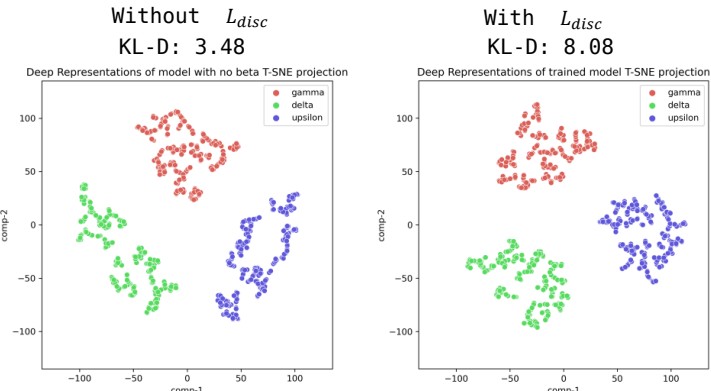

Figure 5: t-SNE plots of the deep representations with/without $L_{disc}$. *KL-D* represents KL-Divergence between three representations.

$w(t_i, \Gamma(x_i), \Delta(x_i))$. We demonstrate their roles by setting the corresponding hyperparameter to 0 while keeping the other hyperparameters fixed at the best-tuned values. Especially, to evaluate the re-weighting function, we set the value of re-weighting to 1. As shown in Table 2, all components contribute to the model performance, as evidenced by the varying extent of performance drop in most scenarios when any part is removed. Moreover, we find that the re-weighting function and the discrepancy loss are two critical components of our model due to the significant increase of test error when they are disabled. In other words, adjusting for selection bias and forcing all representations to encode the corresponding information independently are predominant for the model performance. Additionally, to give a visual demonstration of the contribution of the discrepancy loss, we show the t-SNE plots of the model trained with and without $L_{disc}$ on a synthetic dataset in Fig. 5. After incorporating the discrepancy loss, the distinct representations become more separate, leading to a larger KL-divergence (e.g. 8.08 vs 3.48). However, the re-weighting function contributes less to the News dataset than the other dataset due to the distinct data generation process of News as discussed above. In particular, all features are associated with treatment assignment and outcome generation, which means that all features act as confounders. Consequently, it presents challenges for $\Gamma(x_i)$ and $\Upsilon(x_i)$ in accurately learning instrumental and adjustment factors. Inaccuracies in $\Gamma(x_i)$ and $\Upsilon(x_i)$ can also affect re-weighting function estimation, leading to a limited contribution.

**Sensitivity Analysis.** We have also investigated the effects of different values of $\alpha$, $\beta$, $\gamma$, $\lambda$, and different proportions of the re-weighting values on the model performance. The results shown in Fig. 2 suggest that the re-weighting function and the discrepancy loss ($\beta$) have more substantial influence on the model's performance, which consists with the findings from the previous ablation study. Typically, the figure shows that a relatively large beta value results in better performance. Moreover, the results indicate that DTRNet can learn an accurate re-weighting value to improve performance, as evidenced by the fact that the current proportion (1.0) of re-weighting values yields the lowest error in IDRF estimation. Similarly, the results in terms of AMSE are shown in Fig. 3, and the trend mirrors that of the MISE.

### 4.5 Disentanglement Performance

To answer **Q3**, we attempt to explore whether the representations capture the corresponding factors by utilizing t-SNE to visualize the three deep representations in a synthetic dataset with respect to different types of variables, as shown in Fig. 4. Since we have knowledge of the data generation process of the synthetic dataset, we know the ground truth of which features are instrumental, confounder, and adjustment variables in the dataset. Hence, for each type of variable, we choose one feature to show the relationship between it and the three deep representations. In Fig. 4, the color shade indicates the value of the corresponding variable, while the shape of the points denotes the type of representations. For example, in the first plot, we aim to verify if the representations of instrumental factors embed the information of instrumental variables in the covariates. The plot shows that the color shading changes with the direction of the corresponding arrow in the instrumental representation (dot points), indicating that instrumental representations encode information about the instrumental variable. Specifically, this means that the instrumental representations learn information about the instrumental variables and can distinguish between large and small values of the corresponding variable. However, in other representations, such as $\Delta$ and $\Upsilon$ (cross points and squared points, respectively), the color shade does not change regularly, indicating that the confounder and adjustment representations do not encode information about instrumental variables. Therefore, our DTRNet can decently disentangle the three factors.

## 5 Conclusion

In this paper, we introduce Disentangled Representation Network (DTRNet), a novel model designed to estimate the individualized dose-response function (IDRF) with high precision while accounting for selection bias through disentangled representations of instrumental, confounder, and adjustment factors. Our experiments on synthetic and semi-synthetic datasets show the exceptional disentanglement capabilities of DTRNet and highlight its impressive performance on estimating IDRF, surpassing current SOTA methods.

### 5.1 Limitations and Future Work

Although we propose an advanced model for estimating the unbiased IDRF, there are still areas for improvement. First, our method currently employs a basic density estimator to estimate the treatment effect. This choice was made for proof of concept. Hence, exploring more sophisticated methods could further enhance performance. Similarly, our use of independent loss may not be optimal. Alternative approaches, such as the Hilbert-Schmidt Independence Criterion (HSIC), could potentially improve results. Hence, future work could focus on exploring these alternatives to improve model performance.

Moreover, our study utilizes only synthetic or semi-synthetic datasets, which allows us to generate the ground-truth IDRF using predefined equations. However, in real-world scenarios, it is challenging to obtain the true IDRF and to differentiate between instrumental, confounder, and adjustment features. This complicates the evaluation of the model and its ability to correctly disentangle these features in practical settings. An expert in the relevant domain may be necessary to properly assess the model performance on real-world datasets. Therefore, a key direction for future research is to investigate the deployment of our model in real-world situations.

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

## A   Proofs

**Lemma A.1.** *(Importance Sampling) Let $p(x)$ be a probability density for a random variable $X$ defined on $\mathbb{R}^d$, then for any probability density $q(x) \in \mathbb{R}^d$ that satisfies $q(x) > 0$ whenever $f(x)p(x) \neq 0$, we have:*

$$\mathbb{E}_{x \sim p(x)}[f(x)] = \mathbb{E}_{x \sim q(x)}[f(x)\frac{p(x)}{q(x)}].$$

*Proof.*

$$
\begin{aligned}
\mathbb{E}_{x \sim p(x)}[f(x)] &= \int_{\mathcal{P}} f(x)p(x)dx \\
&= \int_{\mathcal{P}} \frac{f(x)p(x)}{q(x)}q(x)dx \\
&= \mathbb{E}_{x \sim q(x)}[f(x)\frac{p(x)}{q(x)}].
\end{aligned}
\tag{18}
$$

$\square$

**Theorem A.2.** *Let $p(x, t')$, $p(x, t)$ denote the counterfactual and factual probability density for unit $x$, respectively. Under the conditions of the lemma A.1, the expected loss functions for factual and counterfactual outcomes are:*

$$
\begin{aligned}
\epsilon_{CF} &= \mathbb{E}_{x,t \sim p(x,t')}[l_{\Delta,\Upsilon,g_{\theta(t)}}(x,t)] \\
&= \mathbb{E}_{x,t \sim p(x,t)}[l_{\Delta,\Upsilon,g_{\theta(t)}}(x,t)\frac{p(x,t')}{p(x,t)}],
\end{aligned}
\tag{19}
$$

$$
\begin{aligned}
\epsilon_{F} &= \mathbb{E}_{x,t \sim p(x,t)}[l_{\Delta,\Upsilon,g_{\theta(t)}}(x,t)\frac{p(x,t)}{p(x,t)}] \\
&= \mathbb{E}_{x,t \sim p(x,t)}[l_{\Delta,\Upsilon,g_{\theta(t)}}(x,t) \cdot 1],
\end{aligned}
\tag{20}
$$

*where $t'$ represents all counterfactual treatments of $x$, defined as $t' = \{t'|t' \in \mathcal{T}, t' \neq t\}$. The union of $t$ and $t'$ constitutes the universal set of treatments. Therefore, given the probability density function $f(x, t)$ we have,*

$$\mathbb{P}(x,t) + \mathbb{P}(x,t') = \int_t f(x,t)dt + \int_{\mathcal{T}/\{t\}} f(x,t)dt = \int_{\mathcal{T}} f(x,t)dt = \mathbb{P}(x),\tag{21}$$

The theorem implies that the expected loss function for counterfactual outcomes can be derived by reweighting the factual loss. Hence, we can cooperate the counterfactual loss with factual loss and optimise them together through a designed weight (Hassanpour & Greiner, 2019a);

$$w = 1 + \frac{\mathbb{P}(x,t')}{\mathbb{P}(x,t)} = \frac{\mathbb{P}(x)}{\mathbb{P}(x,t)} = \frac{1}{\mathbb{P}(t|x)} = \frac{1}{\mathbb{P}(t|\Gamma(x),\Delta(x))}.\tag{22}$$

The second equality is by Equation 21, the third equality is by the law of conditional probability, the forth equality is by the Assumption 4 and the independency between adjustment factors and treatment.

**Theorem A.3.** *(Bias Removal with Weighted Loss) Under the theorem 1 and all assumptions, we have:*

$$\mathbb{E}[wl_{\Delta,\Upsilon}(x,t)] = \epsilon$$

*Proof.*

$$
\begin{aligned}
\mathbb{E}_{x,t\sim p(x,t)}[wl_{\Delta,\Upsilon}(x,t)] &= \mathbb{E}[(1 + \frac{p(x,t')}{p(x,t)})l_{\Delta,\Upsilon}(x,t)] \\
&= \mathbb{E}_{x,t\sim p(x,t)}[l_{\Delta,\Upsilon}(x,t) + l_{\Delta,\Upsilon}(x,t)\frac{p(x,t')}{p(x,t)})] \\
&= \mathbb{E}_{x,t\sim p(x,t)}[l_{\Delta,\Upsilon}(x,t)] + \mathbb{E}_{x,t\sim p(x,t')}[l_{\Delta,\Upsilon}(x,t)] \\
&= \int_{\mathcal{X}\times\mathcal{T}} l_{\Delta,\Upsilon}(x,t)\mathbb{P}(x,t)dxdt \\
&+ \int_{\mathcal{X}\times\mathcal{T}} l_{\Delta,\Upsilon}(x,t)\mathbb{P}(x,t')dxdt \\
&= \int_{\mathcal{X}\times\mathcal{T}} l_{\Delta,\Upsilon}(x,t)(\mathbb{P}(x))dxdt \\
&= \epsilon
\end{aligned}
\tag{23}
$$

$\square$

This theorem states that the weighted loss is an unbiased estimation of the IDRF loss, which indicates the re-weighting function in our can precisely eliminate selection bias. Notably, each term in our loss function contributes meaningfully to the theoretical proof. Specifically, some components of the loss function serve to support the assumptions underpinning our proof, as outlined in Assumptions 1-4 in our paper. Their functions are introduced as follows. In Assumption 4, we posit that $X$ should be decomposed into three different factors, where discrepency loss $L_{dis}$ is used to ensure this. $L_{ind}$ enforces us to encode all information of $t_i$ in $\Gamma(x_i)$ and $\Delta(x_i)$ instead of $\Upsilon(x_i)$, thereby facilitating a more precise and accurate estimation of the weight $w$. Moreover, the weight $w$ before factual loss is the key of the proof, hence we use the $L_T$ loss to get the accurate weight.

## B  Reason to choose the MISE and AMSE metric

MISE and AMSE are widely used metrics for IDRF evaluation (Zhang et al.; Nie et al., 2021; Bellot et al., 2022; Zhu et al., 2024a). Therefore, we follow these previous works in adopting these metrics as our evaluation metrics. Typically, MISE and AMSE assess the performance of the models by computing the difference between the estimated curve and the ground-truth curve at both the individual and population levels. Given the specific design of our dataset, if a model is trained on the biased dataset and shows low MISE and AMSE on the unbiased evaluation dataset, it indicates that the model is capable of adjusting for selection bias.

## C  Baseline Methods

We compare our model DTRNet with several state-of-the-art methods on continuous treatment effect estimation, including Dragonet, Dragonet_TR (Shi et al., 2019), DRNet, DRNet_TR (Schwab et al., 2020), VCNet, VCNet_TR (Nie et al., 2021), and TransEE (Zhang et al.). TR refers to targeted regularization, a technique used to improve accuracy. The details of baselines are as follows.

- **Dragonet**: (Shi et al., 2019) used a three-headed architecture to predict the propensity score and conditional outcome from covariates and treatment information. The model was later improved by(Nie et al., 2021) by using separate heads for treatment in different intervals to adjust for continuous treatments.

- **DRNet**: (Schwab et al., 2020) proposed to divide continuous treatments into several intervals and assign one head to each interval to generate the dose-respond curve. Following (Nie et al., 2021), DRNet was improved by adding a conditional density estimation head for treatment estimation.

Table 3: Performance comparision between DTRNet and non-neural network baselines

| Method | Synthetic Data | IHDP | News |
|---|---|---|---|
| Causal Forest* [2] | $0.043 \pm 0.0021$ | $0.97 \pm 0.034$ | $0.211 \pm 0.003$ |
| BART* | $0.040 \pm 0.0013$ | $\mathbf{0.33 \pm 0.005}$ | $0.066 \pm 0.003$ |
| GPS* | $0.028 \pm 0.0016$ | $0.67 \pm 0.025$ | $0.022 \pm 0.001$ |
| Ours | $\mathbf{0.013 \pm 0.0070}$ | $0.37 \pm 0.330$ | $\mathbf{0.010 \pm 0.007}$ |

- **Vcnet**: Nie et al. (2021) introduced a varying coefficient structure to allow the prediction head parameters to be functions of continuous treatments.

- **TransTEE**: (Zhang et al.) adopted the transformer backbones to estimate treatment effect.

## D  Performance comparison between DTRNet and non-deep neural network model

We conducted our experiments under identical settings with (Nie et al., 2021), including the same data generation process, dataset, and problem setting. Therefore, we compared our methods against non-deep learning approaches such as causal forest, BART, and GPS (reported by (Nie et al., 2021)) on these three datasets. The results from our experiments consistently indicate that our methods exhibit superior, or at the very least, comparable performance in comparison to these alternative approaches.

## E  Details about the divergence loss

Typically, we do not include the divergence loss between the instrumental representations $\Gamma(x_i)$ and the adjustment representations $\Upsilon(x_i)$. In our model, instrumental $\Gamma(x_i)$ and confounder representations $\Delta(x_i)$ are used to estimate the probability of treatment, while confounder $\Delta(x_i)$ and adjustment representations $\Upsilon(x_i)$ are used to predict the outcome. Hence, instrumental and confounder representations are more similar, as are confounder and adjustment representations, since they are optimized for similar goals. Therefore, we only penalize these. To avoid making our model cumbersome, we choose not to include the divergence loss between instrumental $\Gamma(x_i)$ and adjustment representations $\Upsilon(x_i)$. In our work, we specifically use the implementation in PyTorch to compute $L_D$: KLDivLoss.

## F  Implementation Details and Hyperparameter Tuning

All the neural network-based methods are trained for 800 epochs with the SGD optimizer (momentum = 0.9). To mitigate the risk of overfitting or underfitting, we apply an early stop technique. For the three deep representation networks in our model, we implement them as fully connected networks with two hidden layers, and each layer has 50 hidden units using ReLU activation. We also use two-hidden-layer settings (each with 50 hidden units) for the $Y$ prediction network. We used grid search tuning to tune the following hyperparameters: $\alpha, \beta, \gamma \in \{0.1, 0.2, 0.4, 0.6\}$ and the learning rate (lr)$\in \{0.0001, 0.00005, 0.00001\}$. For other hyperparameters, e.g., the number of knots and the degree of B-spline, we follow the setting of (Nie et al., 2021) that is also tuned on the same configurations of datasets. For each dataset, we generate 50 runs for training and 20 runs for tuning the aforementioned hyperparameters. The best hyperparameter settings are as follows:

For the baselines, we used the best hyperparameters for each dataset as reported by the authors Zhang et al. (2017); Nie et al. (2021). The baseline models were also tuned on the same datasets with 20 runs using grid search. The tuning ranges for Dragonet, DRNet, and VCNet were as follows: learning rate ($lr$) $\in \{0.05, 0.005, 0.001, 0.0005, 0.0001\}$ and $\alpha \in \{1, 0.5\}$. For the TR versions of these models, the learning rate for $\epsilon(t)$ was $lr \in \{0.001, 0.0001\}$, and $\beta \in \{20, 10, 5\} \times n^{\frac{1}{2}}$. For TransTEE, it was also tuned on these three datasets, but the authors only provided the final best hyperparameters used in their paper without specifying the range. Below are the best hyperparameters that we adopted.

Table 4: The best hyperparameters in DTRNet.

|       | Synthetic | News   | IHDP    |
|-------|-----------|--------|---------|
| alpha | 0.6       | 0.6    | 0.6     |
| beta  | 0.2       | 0.2    | 0.6     |
| gamma | 0.6       | 0.6    | 0.1     |
| lr    | 0.00001   | 0.0001 | 0.00005 |

Table 5: The best Hyper-parameters of TransTEE. Bsz indicates the batch size, # Emb indicates the embedding dimension, Lr. S indicates the scheduler of the learning rate (Cos is the cosine annealing Learning rate).

| Dataset | Bsz | # Emb | # Layers | # Heads | Lr     | Lr. S |
|---------|-----|-------|----------|---------|--------|-------|
| Simu    | 500 | 10    | 1        | 2       | 0.01   | Cos   |
| IHDP    | 128 | 10    | 1        | 2       | 0.0005 | Cos   |
| News    | 256 | 10    | 1        | 2       | 0.01   | Cos   |

## G   Dataset Generation

Our generation process is in line with (Nie et al., 2021). The generation process of the three datasets are as follows:

- **Semi-synthetic Data.** We construct 50 synthetic datasets for training and testing our method and baselines. Each dataset contains 500 training samples and 200 test samples. Furthermore, we simulate another 20 datasets to tune the hyperparameters. In this synthetic dataset all features $x \in \mathbb{R}^6$ follow the i.i.d. distribution,

$$
\begin{aligned}
\tilde{t}|x &= \frac{10\sin(\max(x_1, x_2, x_3)) + \max(x_3, x_4, x_5)^3}{1 + (x_1 + x_5)^2} + \sin(0.5x_3)(1 + \exp(x_4 - 0.5x_3)) \\
&\quad + x_3^2 + 2\sin(x_4) + 2x_5 - 6.5 + \mathcal{N}(0, 0.25) \\
y|x,t &= \cos(2\pi(t - 0.5))(t^2 + \frac{4\max(x_1, x_6)^3}{1 + 2x_3^2}\sin(x_4)) + \mathcal{N}(0, 0.25),
\end{aligned}
\tag{24}
$$

where $t = (1 + exp(-\tilde{t}))^{-1}$. According to the above equation, $x_2, x_5$ are instrumental variables, $x_1, x_3, x_4$ are confounder variables, and $x_6$ is the adjustment variable.

- **IHDP.** Infant Health and Development Program (**IHDP**), an RCT dataset, originally compiled to estimate the causal effect of binary treatment (home visits of specialists) on future cognitive test scores. The original dataset contains 747 samples with 25 features, in order to estimate continuous causal effect, we generate the treatment and outcome following (Nie et al., 2021). We randomly split the dataset into training set (67%) and testing set (33%).:

$$
\begin{aligned}
\tilde{t}|x &= \frac{2x_1}{(1 + x_2)} + \frac{2\max(x_3, x_5, x_6)}{0.2 + \min(x_3, x_5, x_6)} + 2\tanh(5\frac{\sum_{i \in S_{dis,2}}(x_i - c_2)}{|S_{dis,2}|}) - 4 + \mathcal{N}(0, 0.25), \\
y|x,t &= \frac{\sin(3\pi t)}{1.2 - t}(\tanh(5\frac{\sum_{i \in S_{dis,2}}(x_i - c_1)}{|S_{dis,1}|}) + \frac{\exp(0.2(x_1 - x_6))}{0.5 + 5\min(x_2, x_3, x_5)}) + \mathcal{N}(0, 0.25),
\end{aligned}
\tag{25}
$$

where $t = (1 + exp(-\tilde{t}))^{-1}$, $S_{con} = \{1, 2, 3, 5, 6\}$ is the index set of continuous features, $S_{dis,1} = \{4, 7, 8, 9, 10, 11, 12, 13, 14, 15\}$, $S_{dis,2} = \{16, 17, 18, 19, 20, 21, 22, 23, 24, 25\}$ and $S_{dis,1} \cup S_{dis,2} = [25] - S_{con}$. Here $c_1 = \mathbb{E}\frac{\sum_{i \in S_{dis,1}} x_i}{|S_{dis,1}|}$, $c_2 = \mathbb{E}\frac{\sum_{i \in S_{dis,2}} x_i}{|S_{dis,2}|}$.

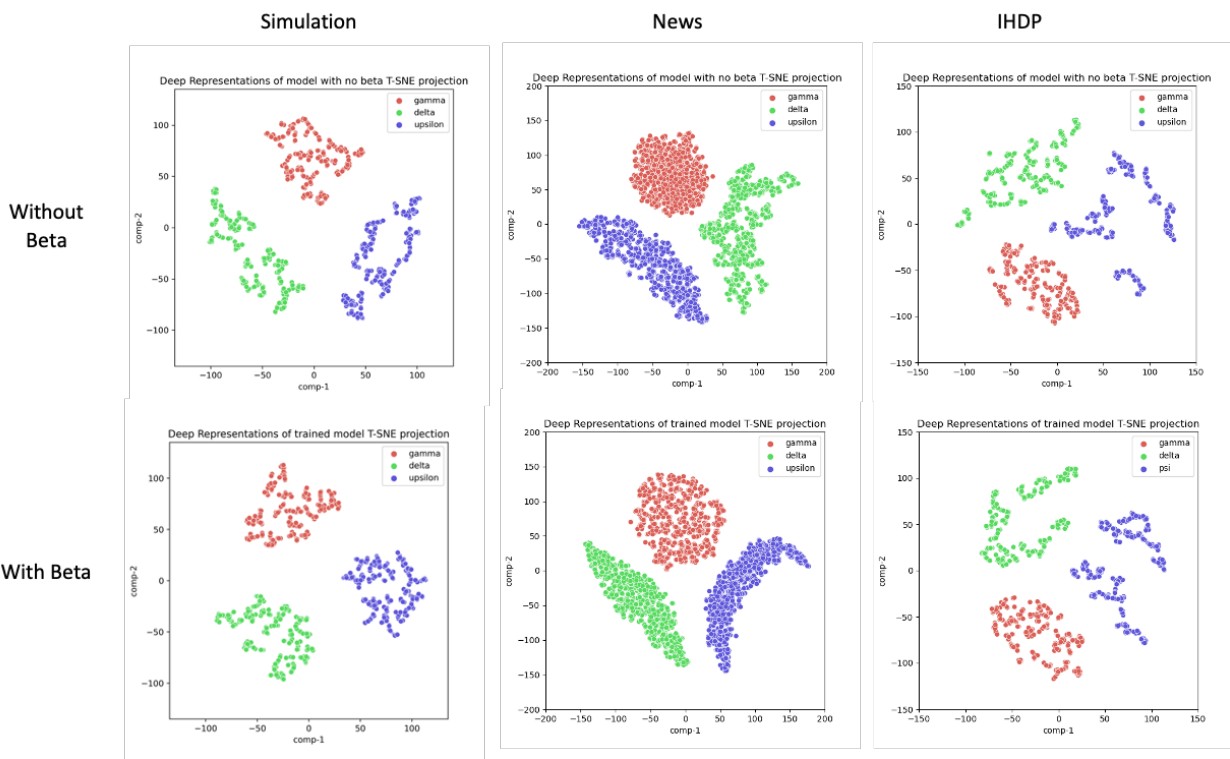

Figure 6: t-SNE plots of the deep represetations with/without $L_{disc}$ across three datasets. For each dataset, we choose one repeat for visualisation due to the space limit.

- **News.** The **News** benchmark is composed of news items from the NY Times corpus, and each item is represented by its word counts. Previous work used it to evaluate the causal effect of devices (binary treatment) on readers' opinions. The original News dataset contains 3000 item samples. Since the dataset is used to estimate the effect of binary treatment, we follow (Nie et al., 2021) to generate continuous treatments as well as the corresponding outcome as follows, we first generate $v'_1, v'_2$ and $v'_3$ from $\mathcal{N}(0, 1)$, and compute $v_i = v'_i/\|v'_i\|_2$. Treatment $t$ is generate from Beta $(2, |\frac{v_3^T x}{2v_2^T x}|)$, where $y$ is generated by:

$$
\begin{aligned}
y'|x, t &= \exp(\frac{v_2^T x}{v_3^T x} - 0.3), \\
y'|x, t &= 2(\max(-2, \min(2, y')) + 20v_1^T x) * (4(t - 0.5)^2 \sin(\frac{\pi}{2}t)) + \mathcal{N}(0, 0.5),
\end{aligned}
\tag{26}
$$

According to the generation algorithm, $X$ and $T$ are highly associated, meaning that the dataset contains confounding bias. We then split this biased dataset into training and testing sets. However, we do not use this biased test set directly for evaluation, as our goal is to estimate the IDRF for each $x$, rather than estimating $y$ for a single $t$ corresponding to $x$. Furthermore, to measure the debiasing capability of the proposed methods, we follow the approach described in (Nie et al., 2021; Zhang et al.). Specifically, to construct the IDRF for each individual, all existing $t$ values in the test set are used to generate the IDRF using $y'|x, t$ along with its $x$ value. This process ensures that there is no association between $x$ and $t$. The average of all $y'$ are used as the ground truth for evaluation. This approach ensures that there is no correlation between $t$ and $x$, as the generation of $t$ no longer relies on $t|x$, thereby eliminating selection bias and allowing us to test on an unbiased dataset.

## H  Discrepancy loss on other dataset

In addition, to give a visual demonstration of the contribution of the discrepancy loss, we show the t-SNE plots of models trained with and without $L_{disc}$ on each dataset in Fig. 6. Points belonging to different factors (i.e., $\{\Gamma(x_i), \Delta(x_i), \text{and} \Upsilon(x_i)\}$) are further away under the model trained with the discrepancy loss, indicating that the discrepancy loss indeed helps learn separable representations. Moreover, except for the News dataset, points belonging to the same factor are closer together with $L_{disc}$, suggesting that they extract similar information. However, $\Gamma(x_i)$ in the News dataset becomes more separate, and $\Upsilon(x_i)$ gathers less close compared to other datasets, indicating that it cannot well extract the corresponding information, which is consistent with the explanation of re-weighting above.

## I  An instantiation of causal graph

In this section, we provide an example of the instantiation of our causal graph. Similar to (Wu et al., 2020): In the context of medical health record, we might collect extensive historical data from each patient, including the patients' features $X$ (e.g., age, gender, living environment, doctor-in-charge), treatment of patients $T$ (taking a particular mediciine or not), and the final outcome $Y$ (cured or not). Among these features, age and gender simultaneously affect the treatment (as a physician would consider these factors when choosing a treatment) and the outcome (since they can also affect the patient's recovery rate); therefore, they are confounding factors $\Delta$. In contrast, the doctor-in-charge would influence only the treatment decision, without affecting the outcome, thus being an instrumental factor $\Gamma$. The environment (e.g. hygiene situation), which only affects the outcome but not the treatment, falls into the category of adjustment factors $\Upsilon$.

## J  Sensitivity Analysis based on AMSE

In this section, we present a sensitivity analysis based on AMSE, as shown in Figure 3. The trend observed is similar to that of the MISE, which is presented in the main text. This analysis demonstrates that the re-weighting function and the discrepancy loss $\beta$ have a more substantial influence on the model's performance. These findings are consistent with those from our previous ablation study.

