# OpenReview forum: "DTRNet: Precisely Correcting Selection Bias in Individual-Level Continuous Treatment Effect Estimation by Reweighted Disentangled Representation"
_TMLR — Accepted by TMLR_

### Review · Reviewer_iCdT · 2024-08-25

**Summary Of Contributions:**

The paper proposes Disentangled Representation Network (DTRNet) to precisely estimate the individual dose-response function (IDRF) with continuous treatments and adjust for selection bias. Specifically, the authors use 3 contracted neural networks to learn separate representations for instrumental variables, confounders, and adjustment variables. These representations are then combined in different ways to predict the treatment $T$ and the outcome $Y$, and to encourage independence. In addition, the authors provide a theoretical proof on how the re-weighted expected loss eliminates selection bias using the basic concept of importance sampling. Finally, DTRNet is tested on synthetic and semi-synthetic datasets to evaluate its accuracy in estimating IDRF and reducing selection bias. Ablation studies are also performed to demonstrate the effectiveness of each component in the loss function, especially the re-weighting and the discrepancy loss terms.

**Audience:**

Yes

**Broader Impact Concerns:**

I do not foresee any ethical concerns of this work, and a Broader Impact statement section may not be necessary at this point.

**Claims And Evidence:**

No

**Requested Changes:**

•	The imbalanced regularization term $L_{imb}$ in Figure 1(b) is never used in the actual loss function in Eq. 3. Should it be $L_{ind}$ instead?

•	Should it be $Y(T = t) \perp T | X$ instead of $Y(T = t) \perp t | X$ in Assumption 3.3?

•	What is the factual loss for comparing the ground truth with the estimated targets in Eq. 4, mean-squared error?

•	What is the input $w$ in $\pi^{NN}(\Gamma(x_i), \Delta(x_i)) = softmax(w, \Gamma(x_i), \Delta(x_i))$?

•	There’s a typo in Eq. 12. It should be $\epsilon_F = \mathbb{E}\_{x,t \sim p(x,t)}$.

•	In Section 2.2, additional citations can be included for estimating ITE with machine learning [1] and deep representation learning [2-4] methods.

•	In Section 3.2 – Independent loss, there should be relevant citations when the authors argue that instrumental factors should not be balanced according to the causal theory.

References:

[1]. Alaa, A. M., & Van Der Schaar, M. (2017). Bayesian inference of individualized treatment effects using multi-task gaussian processes. Advances in neural information processing systems, 30.

[2]. Louizos, C., Shalit, U., Mooij, J. M., Sontag, D., Zemel, R., & Welling, M. (2017). Causal effect inference with deep latent-variable models. Advances in neural information processing systems, 30.

[3]. Yoon, J., Jordon, J., & Van Der Schaar, M. (2018, February). GANITE: Estimation of individualized treatment effects using generative adversarial nets. In International conference on learning representations.

[4]. Jiang, Z., Hou, Z., Liu, Y., Ren, Y., Li, K., & Carlson, D. (2023, July). Estimating causal effects using a multi-task deep ensemble. In International Conference on Machine Learning (pp. 15023-15040). PMLR.

**Strengths And Weaknesses:**

**Strengths**:

•	Causal effect estimation with continuous treatment is a relatively underexplored area of research and the problem being addressed by the paper is important.

•	The background and methodology parts of the paper are well written and easy to understand in general.

•	The experimental results demonstrate that the model effectively captures the IDRF, outperforming other state-of-the-art methods.

•	The ablation study is valuable in highlighting the necessity of the different terms in the loss function.

**Weaknesses**:

•	Based on Equation 3 and the experimental setup described in Section 4.2, there are several hyperparameters that require tuning, including the weighting of each loss term and the architecture of the neural networks used for extracting deep representations. While the authors state that they employed grid search to identify the optimal combination of these hyperparameters, it is unclear how the parameter ranges were determined. Further clarification on this point would be beneficial. Additionally, it would be useful to know whether the same fine-tuning strategy was applied to the benchmark methods when generating the results presented in Table 1.

•	In Section 4.3, it makes sense to use the MISE and AMSE metrics for evaluating how well the model captures the IDRF. However, I don’t think these metrics directly demonstrate the model’s capability in adjusting for the selection bias. Since the experiments mainly rely on synthetic and semi-synthetic datasets, is it possible to calculate the bias, i.e., the difference between the ground truth expected IDRF and the re-weighted expected loss?

•	Figure 4 seems hard to understand. In particular, why does the fact that the shade of the color varies along a certain direction indicates disentanglement? I think a more intuitive visualization is to show the distribution of instrumental, confounder, and adjustment factors with and without the discrepancy loss $L_{ind}$.

---

> ### Author Response · Authors · 2024-08-27
> **Author Respond -1**
>
> Thank you for your insightful suggestions.
>
> For Requested Changes
>
> We are grateful for your detailed requested changes, which have significantly improved the paper. We have made **ALL** the requested changes in our latest version and highlighted them in blue. We have corrected the typos you identified, both in the figure and the text and added the corresponding citations for estimating ITE with machine learning and deep representation learning, as well as for our claim regarding Independent Loss. Regarding the two questions from your requested changes:
>
> (1) What is the factual loss for comparing the ground truth with the estimated targets in Eq. 4, mean-squared error?
>
> Yes, it refers to mean squared error. Thank you for pointing that out; we have clarified this in the latest version.
>
> (2)  What is the input $w$ in $\pi^{NN}(\Gamma(x_i),\Delta(x_i)) = softmax(w, \Gamma(x_i),\Delta(x_i)) \in \mathbb{R}^{B+1}$?
>
>  It actually a typo, the correct equation would be $\pi^{NN}(\Gamma(x_i),\Delta(x_i)) = softmax( \textbf{W}_t\cdot concat(\Gamma(x_i),\Delta(x_i))) \in \mathbb{R}^{B+1}$, where $\textbf{W}_t$ is the parameter for the network. We have corrected it, thanks for pointing it out!

---

> ### Author Response · Authors · 2024-08-27
> **Author Respond -2**
>
> For the Weakness
>
>  Q1: The choice of hyperparameter and the motivation for the hyperparameter tuning range.
>
> Thank you for bringing this to our attention. We indeed had not included this in our paper. In response to your suggestion, **we have added this part to the implementation details and highlighted it in blue** in Appendix E. Here is the motivation:
>
>  The choice of the model structure aligns with previous approaches[1,2]. We use grid search tuning to tune the following hyperparameters: $\alpha,\beta,\gamma\in{0.1,0.2,0.4,0.6}$. The choice of this range is based on the common practice of weighting loss terms between 0 and 1. Specifically, in our paper, these hyperparameters refer to the weighting of each loss term, except for the factual loss (the loss between $y_i$ and $\hat{y_i}$). Since the factual loss represents the loss of prediction $y$, which is a crucial component of the total loss, assigning a high weight to other terms could diminish the impact of the factual loss. Therefore, we chose the range ${0.1, 0.2, 0.4, 0.6}$, also considering computational and time costs.
>
>   For the baselines, yes, the same fine-tuning strategy was applied.  Specifically, we used the best hyperparameters reported by the authors [1,2] for each dataset. The baseline models were also tuned on the same datasets with 20 runs using the same fine-tuning strategy. The tuning ranges for Dragonet, DRNet, and VCNet were as follows: learning rate ($lr$) $\in {0.05, 0.005, 0.001, 0.0005, 0.0001}$ and $\alpha \in {1, 0.5}$. For the TR versions of these models, the learning rate for $\epsilon(t)$ was $lr \in {0.001, 0.0001}$, and $\beta \in {20, 10, 5} \times n^{\frac{1}{2}}$. For TransTEE, it was also tuned on these three datasets, but the authors only provided the final best hyperparameters used in their paper without specifying the range. Below are the best hyperparameters that we adopted.
>
> | Dataset | Bsz | \# Emb | \# Layers | \# Heads | Lr   | Lr. S |
> |---------|-----|--------|-----------|----------|------|-------|
> | Simu    | 500 | 10     | 1         | 2        | 0.01 | Cos   |
> | IHDP    | 128 | 10     | 1         | 2        | 0.0005 | Cos   |
> | News    | 256 | 10     | 1         | 2        | 0.01 | Cos   |
>
>
>  Q2. Clarification on the debias ability of our metrics and dataset.
>
> Thank you for raising this concern. We appreciate the opportunity to clarify. Unlike typical datasets where both training and test datasets may be biased, our dataset follows the design principles outlined in prior work [1,2,3]. Specifically, all training sets intentionally contain selection bias, while the evaluation ground truth remains unbiased. For further clarity, we have added details about the generation process of the training and evaluation sets in the new Appendix F of the latest version of our paper.
>
> In this context, a model trained on a biased dataset without the capability to adjust for bias is likely to perform poorly on an unbiased evaluation dataset. Conversely, a model like our DTRNet, which is trained on a biased dataset and can effectively adjust for this bias, should demonstrate good performance on an unbiased dataset. This is because our model can successfully remove the selection bias. Additionally, MISE and AMSE are commonly used metrics for such evaluations [1-4].
>
> Regarding your suggestion, we find the idea of calculating bias interesting. However, even if we could calculate the selection bias, determining the ground truth selection bias is challenging. While we generate the data, there is no established equation for calculating this bias. As a result, comparing the computed bias with the true bias for model evaluation is difficult. Furthermore, since we focus on reweighting the loss rather than the prediction (unlike inverse propensity score methods), the difference between the ground truth IDRF (biased prediction) and the reweighted loss (reweighted prediction difference instead of reweighted prediction) may not directly reflect the selection bias. Therefore, we believe that MISE and AMSE are more suitable for evaluating our model in this situation. However, we find your idea interesting and believe it is worth exploring further. Developing a new evaluation metric based on this concept could be a valuable direction for future work.

---

> ### Author Response · Authors · 2024-08-27
> **Author Respond -3**
>
> 3. Clarification on the Figure 4
>
> We apologies that this brings you the confusion. We here provide further clarifications, this figure is not intended to illustrate the function of discrepancy loss. Instead, our goal is to investigate whether the representations accurately capture the corresponding factors. For example, in the first plot of Figure 4, we aim to determine if the representations for instrumental factors (the group of dot points in Figure 4) truly embed information about instrumental variables in the covariates (since we know which features are instrumental, confounder, or adjustment variables based on the data generation process of the synthetic dataset), while not embedding information about confounder and adjustment variables. To demonstrate this, we use the shape of the points to represent different embeddings and the color shading to represent the value of the instrumental variable in the covariate. The plot shows that the color shading changes with the direction of the corresponding arrow in the instrumental representation (dot points), indicating that instrumental representations encode information about the instrumental variable. Specifically, this means that the instrumental representations learn information about the instrumental variables and can distinguish between large and small values of the corresponding variable. However, other representations, such as confounder representations or adjustment representations (cross points/square points), do not show a regular change in shading, indicating that they do not encode information about instrumental variables and cannot distinguish between large and small values. Each plot is designed to assess whether the corresponding representation captures a specific variable: plot 2 examines if the confounder representations capture information about confounder variables while other representations do not, and plot 3 examines if the adjustment representations capture information about adjustment variables while other representations do not.
>
> Thank you for pointing this out; **we have added more clarifications in this section**.
>
> We hope that our response addresses your concerns adequately, and we are committed to incorporating your valuable feedback into the revised version of our paper. Thank you once again for your insightful comments.
>
>  [1]Zhang Y F, Zhang H, Lipton Z C, et al. Exploring transformer backbones for heterogeneous treatment effect estimation[J]. TMLR 2024
>
>  [2] Nie L, Ye M, Nicolae D. VCNet and Functional Targeted Regularization For Learning Causal Effects of Continuous Treatments[C]//International Conference on Learning Representations.
>
>  [3]Bellot A, Dhir A, Prando G. Generalization bounds and algorithms for estimating conditional average treatment effect of dosage[J]. arXiv preprint arXiv:2205.14692, 2022.
>
>  [4]Zhu M, Wu A, Li H, et al. Contrastive balancing representation learning for heterogeneous dose-response curves estimation[C]//Proceedings of the AAAI Conference on Artificial Intelligence. 2024, 38(15): 17175-17183.

---

> ### Comment · Reviewer_iCdT · 2024-09-17
> **Response to rebuttal**
>
> Thank you for your detailed clarifications and responses to my questions. I think most of my concerns are addressed by the rebuttal. Regarding the debiasing metrics, I think it would be helpful to add the discussion on why MISE and AMSE are preferred over other metrics. Also, for Figure 4, I think the authors should clarify what the arrow in each subfigure represents in both the figure caption and the texts. For the rightmost subfigure (i.e., adjustment variable), it seems that the color shading of both $\Delta$ and $\Upsilon$ representations change with the arrow direction. Does this mean $\Delta$ and $\Upsilon$ learn from the adjustment variable? Is this a desired behavior?

---

> > ### Author Response · Authors · 2024-09-19
> > **Further Clarifications**
> >
> > Dear Reviewer iCdT,
> >
> > Thank you for your responses and further suggestions, we believe they are very beneficial for improving our paper.
> >
> > Regarding the evaluation metrics we adopted, **we have added the reasons for choosing these metrics in both Section 4.3 and the detailed explanation in Appendix B** (highlighted in blue). MISE and AMSE are widely used metrics for IDRF evaluation [1-4]. Therefore, we follow these previous works in adopting these metrics as our evaluation metrics. Typically, MISE and AMSE assess the performance of the models by computing the difference between the estimated curve and the ground-truth curve at both the individual and population levels. Given the specific design of our dataset, if a model is trained on the biased dataset and shows low MISE and AMSE on the unbiased evaluation dataset, it indicates that the model is capable of adjusting for selection bias.
> >
> > Thank you for pointing this out; **we have added clarifications regarding the arrows in Figure 4 and highlighted them in blue**. These arrows are manually added to the figures to provide readers with a clear visual indication of the directions in which the color changes in the corresponding embeddings. Regarding the rightmost figure, we kindly argue that for $\Upsilon$, the color changes gradually (from dark to light) according to the arrow direction. In contrast, for $\Delta$, both the bottom and middle sections contain dark blue points, while a dark point is also located at the top of the embedding. Hence, we believe that $\Delta$ does not properly encode information about the adjustment variable.
> >
> > We hope that our response adequately addresses your concerns, and we are committed to incorporating your valuable feedback into the revised version of our paper. Thank you once again for your insightful comments.
> >
> > [1]Zhang Y F, Zhang H, Lipton Z C, et al. Exploring transformer backbones for heterogeneous treatment effect estimation[J]. TMLR 2024
> >
> > [2] Nie L, Ye M, Nicolae D. VCNet and Functional Targeted Regularization For Learning Causal Effects of Continuous Treatments[C]//International Conference on Learning Representations.
> >
> > [3]Bellot A, Dhir A, Prando G. Generalization bounds and algorithms for estimating conditional average treatment effect of dosage[J]. arXiv preprint arXiv:2205.14692, 2022.
> >
> > [4]Zhu M, Wu A, Li H, et al. Contrastive balancing representation learning for heterogeneous dose-response curves estimation[C]//Proceedings of the AAAI Conference on Artificial Intelligence. 2024, 38(15): 17175-17183.

---

> ### Author Response · Authors · 2024-10-05
> **Response to Review Feedback**
>
> Dear reviewer iCdT,
>
> Please allow us to thank you again for reviewing our paper and for the valuable feedback, particularly for recognizing the strengths of our paper in terms of the problem, writing, experiments, and theoretical proofs.
>
> Kindly let us know if our response has properly addressed your concerns. We are more than happy to answer any additional questions during the discussion period. Your feedback will be greatly appreciated.
>
> Best,
> Authors

---

### Review · Reviewer_YgC7 · 2024-09-04

**Summary Of Contributions:**

This paper proposes a way to adjust for selection bias under continuous settings. Disentangled Representation Network(DTRNet) learns disentangled representations and adjusts for selection bias, by estimating the individualized dose-response function (IDRF). The paper also includes some theory proofs for the effectiveness of DTRNet. By experiments in this paper, the authors evaluate their methods in three ways: 1. The effectiveness of DTRNet in estimating IDRF and adjusting for selection bias compared to the state-of-art methods. 2. The individual contributions of each complement in the model. 3. The effectiveness of the deep entangled representations in identifying the three underlying factors.

**Audience:**

Yes

**Claims And Evidence:**

Yes

**Requested Changes:**

1. I have this question and hope the authors can answer: One assumption for the theoretical proof is that there are no interactions between units and there is only one version of each treatment. But how realistic is it in the case of continuous settings, especially not by dividing the continuous treatments into bins?
2. Like mentioned above, it would be more convincing to add comparison with other state-of-art casual learning methods also via neural networks, like the “circe” method in “Efficient Conditionally Invariant Representation Learning”.

**Strengths And Weaknesses:**

Strengths:
1. It is solid to include theoretical proofs for the effectiveness of Disentangled Representation Network for disentangled representations learning. The proofs are easy to follow.
2. The experiments cover three important questions analyzing the different perspectives about the adjustment for selection bias.

Weakness:
1. The baseline methods do not contain many state-of-art casual learning methods also via neural networks. For example, like methods in this paper “Efficient Conditionally Invariant Representation Learning”. Also not sure how the biased training dataset and unbiased testing dataset is divided and whether it’s fair.

---

> ### Author Response · Authors · 2024-09-10
> **Author Respond -1**
>
> Requested changes:
>
> We are grateful for your detailed requested changes, which have significantly improved the paper. We have made **ALL** the requested changes in our latest version and highlighted them in blue.
>
> 1. **Clarifications of SUTVA assumption.**
>
>   Thank you for pointing this out. This expression indeed introduces confusion regarding the assumption, and we appreciate the opportunity to clarify and revise it. The Stable Unit Treatment Value Assumption (SUTVA) can be broken down into two conditions: no interference and well-defined treatment levels[1-2].
>
>   No interference signifies that the potential outcomes of an instance are independent of the treatments received by other units. For example, the outcome of one patient (A) should not be affected by the treatment that another patient (B) receives.
>
>   In the continuous setting, well-defined treatment levels indicates that if two different instances $i$ and $j$ have the same value for their treatment variable, then they receive the same treatment [3]. For example, if both patient A and patient B are receiving the same medicine at an amount $c$ (where $c$ is a continuous value), then we consider them to be receiving the same treatment.
>
>   We apologize for the confusion caused by using an example with binary treatments (there is only one version of each treatment and different levels or doses of a treatment are treated as different treatments) instead of continuous treatments to explain this well-defined treatment levels.  This example is indeed not suitable for this paper. Hence **we have revised this part and highlighted it in blue**.
>
> [1]Yao L, Chu Z, Li S, et al. A survey on causal inference[J]. ACM Transactions on Knowledge Discovery from Data (TKDD), 2021, 15(5): 1-46.
>
> [2]Guo R, Cheng L, Li J, et al. A survey of learning causality with data: Problems and methods[J]. ACM Computing Surveys (CSUR), 2020, 53(4): 1-37.
>
> [3]Zeng Z, Arbour D, Feller A, et al. Continuous Treatment Effects with Surrogate Outcomes[J]. arXiv preprint arXiv:2402.00168, 2024.

---

> ### Author Response · Authors · 2024-09-10
> **Author Respond -2**
>
> 2. **Add Advanced Baseline.**
> Thank you for providing us with this interesting paper for comparison. We have included this baseline in our study.
>
> The paper introduces CIRCE as a regularizer in settings where $X$ is used to predict $Y$. Specifically, the regularizer aims to learn neural features $\phi(X)$ of the data $X$ such that $\phi(X)$ is independent of some metadata $Z$ given $Y$. However, since CIRCE was not originally proposed for treatment effect estimation, its evaluations focused on two tasks: (1) synthetic data of moderate dimension to study the effectiveness of CIRCE at enforcing conditional independence under established settings (e.g., econometrics or epidemiology), and (2) high-dimensional image data to learn image representations robust to domain shifts. Therefore, CIRCE cannot be directly applied to our task. Hence, to adapt this approach to our setting, where we aim to make $X$ independent of $T$ given $Y$, we use a classic method, Dr_Net, as the backbone for this adaptation. We have compared this adaptation with our model, and the results are as follows.
>
> | **Method**      | **Synthetic Data (MISE)** | **Synthetic Data (AMSE)** | **News (MISE)** | **News (AMSE)** | **IHDP (MISE)** | **IHDP (AMSE)** |
> | --------------- | ------------------------- | --------------------------| --------------- | --------------- | --------------- | --------------- |
> | Dragonet        | 0.1854 ± 0.0232           | 0.0415 ± 0.0081           | 1.3241 ± 0.1617 | 0.0535 ± 0.0053 | 4.7034 ± 0.5860 | 0.9549 ± 0.3005 |
> | Dragonet_TR     | 0.1720 ± 0.0219           | 0.0281 ± 0.0095           | 1.3147 ± 0.1594 | 0.0401 ± 0.0062 | 4.2877 ± 0.4226 | 0.6490 ± 0.1660 |
> | DRNet           | 0.1849 ± 0.0232           | 0.0409 ± 0.0081           | 1.3248 ± 0.1616 | 0.0542 ± 0.0054 | 4.7394 ± 0.6036 | 0.9581 ± 0.3324 |
> | DRNet_TR        | 0.1752 ± 0.0334           | 0.0315 ± 0.0235           | 1.3148 ± 0.1601 | 0.0403 ± 0.0060 | 4.1313 ± 0.6320 | 0.6140 ± 0.1954 |
> | VCNet           | 0.1545 ± 0.0248           | 0.0173 ± 0.0093           | 2.3372 ± 0.1808 | 0.0384 ± 0.0367 | 3.6651 ± 0.6409 | 0.6755 ± 0.4875 |
> | VCNet_TR        | 0.1418 ± 0.0299           | 0.0142 ± 0.0072           | 2.3289 ± 0.2009 | 0.0378 ± 0.0401 | 3.7935 ± 1.3625 | 1.2302 ± 1.2198 |
> | TransTEE        | 0.2033 ± 0.0978           | 0.0552 ± 0.0884           | **1.2849 ± 0.1587** | 0.0153 ± 0.0066 | 4.1562 ± 0.8053 | 1.8529 ± 1.1155 |
> | **CIRCE[4]**       | 0.6854 ± 0.4050           | 0.5406 ± 0.4040           | 1.7667 ± 0.1995 | 0.6030 ± 0.0291 | 10.4413 ± 5.7308 | 8.218 ± 5.4327 |
> | **DTRNet (Ours)**| **0.1414 ± 0.0256**       | **0.0131 ± 0.0072**       | 1.7846 ± 0.2202 | **0.0104 ± 0.0044** | **3.5376 ± 0.5285** | **0.4254 ± 0.3710** |
>
>
> It shows that this method does not outperform ours. We believe the reasons are as follows:
>
> 1. To adjust selection bias, causal methods[1-3] aim to make $X$ **marginally independent** of $T$ (i.e., the embedding of $X$ ($\phi(X)$) should be independent of $T$). However, CIRCE enforces **conditional independence** (i.e., $\phi(X)$ should be independent of $T$ given $Y$), which is a more relaxed condition than marginal independence. Consequently, the extent of bias adjustment may not be as effective as in other causal models, leading to a larger loss.
>
> 2. CIRCE requires an additional held-out set for ridge regression from $Y$ to the kernelized features of $Z$. Therefore, for small datasets, the accuracy of this estimation is limited. We have also included this paper in the related work section for further discussion.
>
> [1]Zhu M, Wu A, Li H, et al. Contrastive balancing representation learning for heterogeneous dose-response curves estimation[C]//Proceedings of the AAAI Conference on Artificial Intelligence. 2024, 38(15): 17175-17183.
>
> [2] Shalit U, Johansson F D, Sontag D. Estimating individual treatment effect: generalization bounds and algorithms[C]//International conference on machine learning. PMLR, 2017: 3076-3085.
>
> [3] Bellot A, Dhir A, Prando G. Generalization bounds and algorithms for estimating conditional average treatment effect of dosage[J]. arXiv preprint arXiv:2205.14692, 2022.
>
> [4] Pogodin R, Deka N, Li Y, et al. Efficient conditionally invariant representation learning[J]. arXiv preprint arXiv:2212.08645, 2022.

---

> ### Author Response · Authors · 2024-09-10
> **Author Respond -3**
>
> 3. **Dataset Division**
>
> Thank you for raising this concern. **We have added the detailed generation algorithm in Appendix F**.
>
> To construct the dataset, we follow the methodology outlined in [1-2], where each $x$ is generated with one $t$ and one $y$. According to the generation algorithm, $x$ and $t$ are highly associated, meaning that the dataset contains confounding bias. We then split this biased dataset into train and test sets. However, we do not use this biased test set directly for evaluation, as our goal is to estimate the IDRF for each $x$, rather than estimating $y$ for a single $t$ corresponding to $x$.
>
> Specifically, to construct the IDRF for each individual, we use all existing $t$ values in the test set to generate the IDRF using $y'|x,t$ along with each $x$ value. This process ensures that there is no association between $x$ and $t$. The average of all $y'$ values is used as the ground truth for evaluation. This approach guarantees that there is no correlation between $t$ and $x$, as the generation of $t$ no longer depends on $x$, thereby eliminating selection bias and allowing us to test on an unbiased dataset [1-2].
>
>  [1]Zhang Y F, Zhang H, Lipton Z C, et al. Exploring transformer backbones for heterogeneous treatment effect estimation[J]. TMLR 2024
>
>  [2] Nie L, Ye M, Nicolae D. VCNet and Functional Targeted Regularization For Learning Causal Effects of Continuous Treatments[C]//International Conference on Learning Representations.
>
> We hope that our response addresses your concerns adequately, and we are committed to incorporating your valuable feedback into the revised version of our paper. Thank you once again for your insightful comment.

---

> ### Author Response · Authors · 2024-10-05
> **Response to Review Feedback**
>
> Dear reviewer YgC7,
>
> Please allow us to thank you again for reviewing our paper and for the valuable feedback, particularly for recognizing the strengths of our paper in terms of experiments, and theoretical proofs.
>
> Kindly let us know if our response has properly addressed your concerns. We are more than happy to answer any additional questions during the discussion period. Your feedback will be greatly appreciated.
>
> Best,
> Authors

---

> > ### Comment · Reviewer_YgC7 · 2024-10-21
> >
> > Thanks for your response and that has solved my questions.

---

### Review · Reviewer_s1mR · 2024-09-23

**Summary Of Contributions:**

The paper proposes DTRNet, a novel model for estimating the individualized dose-response function (IDRF) under continuous treatment settings. DTRNet introduces disentangled representations of instrumental, confounder, and adjustment factors to address selection bias.
Theoretical proofs support the model's ability to eliminate bias. Experiments on synthetic and semi-synthetic data show that DTRNet outperforms existing state-of-the-art models.

**Audience:**

Yes

**Claims And Evidence:**

Yes

**Requested Changes:**

None

**Strengths And Weaknesses:**

Strengths:
Novelty: First to tackle selection bias for continuous treatments via disentangled representations.
Comprehensive approach: Rigorous theoretical grounding with clear contributions on disentangling confounders from instrumental factors.
Solid experiments: Extensive tests with synthetic/semi-synthetic datasets showing superiority over competing models.

Weaknesses:
Real-world applicability: Lack of testing on real-world datasets limits the model’s practical validation.
Basic density estimator: The use of a simple density estimator may hinder performance.
Independent loss design: Current approach may not be optimal; more advanced methods like HSIC should be explored.

---

> ### Author Response · Authors · 2024-09-25
> **Author Respond**
>
> Dear Reviewer s1mR,
>
> We sincerely appreciate your thoughtful review and thank you for acknowledging the strengths of our paper.
> We completely agree with your observations regarding the paper's weaknesses, which we have discussed in the limitations and future work section (5.1) of the first draft. We believe that dedicated efforts will be essential to address these issues in future research.
>
> 1. No Real-world Data: This limitation is not only related to IDRF estimation but is also a broader challenge within causal inference research. The lack of ground truth for treatment effects in real-world data arises because collecting such data can be expensive or even unethical. For instance, it would be unethical to assign all possible dosages of a treatment to patients solely for the purpose of determining the accurate treatment effect, rather than providing them with the optimal dosage. Therefore, it is common practice in the field of causal inference to utilize synthetic and semi-synthetic data [1-4].
> 2. Basic Density Estimator and Independent Loss Design: Our current method employs a basic density estimator and a straightforward independent loss design, which were chosen for proof of concept. Our experiments indicate that this simple approach can achieve advanced performance. While these methods are both easy to understand and effective, we acknowledge that exploring more sophisticated techniques could further enhance performance.
>
> We hope that our response adequately addresses your concerns, and we are committed to incorporating your valuable feedback into the revised version of our paper. Thank you once again for your insightful comments.
>
> [1]Zhang Y F, Zhang H, Lipton Z C, et al. Exploring transformer backbones for heterogeneous treatment effect estimation[J]. TMLR 2024
>
> [2] Nie L, Ye M, Nicolae D. VCNet and Functional Targeted Regularization For Learning Causal Effects of Continuous Treatments[C]//International Conference on Learning Representations.
>
> [3]Bellot A, Dhir A, Prando G. Generalization bounds and algorithms for estimating conditional average treatment effect of dosage[J]. arXiv preprint arXiv:2205.14692, 2022.
>
> [4]Yao L, Chu Z, Li S, et al. A survey on causal inference[J]. ACM Transactions on Knowledge Discovery from Data (TKDD), 2021, 15(5): 1-46.

---

> ### Author Response · Authors · 2024-10-05
> **Response to Review Feedback**
>
> Dear reviewer s1mR,
>
> Please allow us to thank you again for reviewing our paper and for the valuable feedback, particularly for recognizing the strengths of our paper in terms of novelty, experiments, and theoretical proofs.
>
> Kindly let us know if our response has properly addressed your concerns. We are more than happy to answer any additional questions during the discussion period. Your feedback will be greatly appreciated.
>
> Best,
> Authors

---

### Decision · Action_Editor_TLzx · 2024-11-07

**Recommendation:** Accept as is

**Comment:**

The paper clearly meets the publication criteria of TMLR. It makes a concrete contribution for the literature and it otherwise conducted with good scientific rigour, and despite somewhat limited novelty compared to previous works it makes a clear contribution for the literature on this topic. The manuscript was clearly improved during the review process, for instance expanding the empirical evaluation and providing additional information and clarifications regarding the method. There are no open issues that would need to be resolved.

**Audience:**

The problem is of interest for a broad audience, including both ML researchers working on the problem as well end-users in need of ITE estimates.

**Claims And Evidence:**

The paper proposes a new computational method for estimating individual treatment effects for continuous treatments, such that selection bias can be adjusted separately. The method is claimed to be the first of this kind, which appears correct, and its properties are validated with a theoretical proof of bias elimination and empirical experiments that confirm the comparative claim of outperforming previous methods. The comparison methods include examples of fairly recent methods, including one that was added during the revision.

Overall, the claims are clear and supported by clear evidence.